# Compilation of a database of Holocene nearshore marine mollusc shell geochemistry from the California Current System

Hannah M. Palmer[1], Veronica Padilla Vriesman[1], Roxanne M. W. Banker[2], Jessica R. Bean[3]

1. Earth and Planetary Sciences, University of California, Davis, Davis, CA
2. California Academy of Sciences, San Francisco, CA
3. Museum of Paleontology, University of California, Berkeley, Berkeley, CA

*Correspondence to:* Hannah M. Palmer (hmpalmer@ucdavis.edu)

**Abstract**

The shells of marine invertebrates can serve as high-resolution records of oceanographic and atmospheric change through time. In particular, oxygen and carbon isotope analyses of nearshore marine calcifiers that grow by accretion over their lifespans provide seasonal records of environmental and oceanographic conditions. Archaeological shell middens generated by Indigenous communities along the Northwest coast of North America contain shells harvested over multiple seasons for millennia. These shell middens, as well as analyses of archival and modern shells, have the potential to provide multi-site, seasonal archives of nearshore conditions throughout the Holocene. A significant volume of oxygen and carbon isotope data from archaeological shells exists, yet is separately published in archaeological, geochemical, and paleoceanographic journals and has not been comprehensively analyzed to examine oceanographic change over time. Here, we compiled a database of previously published oxygen and carbon isotope data from archaeological, archival, and modern marine molluscs from the California Current System (North American coast of the Northeast Pacific, 32ºN to 55ºN). This database includes oxygen and carbon isotope data from 598 modern, archaeological, and sub-fossil shells from 8880 years before present (BP) to the present, from which there are 4,917 total $\delta^{13}$C and 7,366 total $\delta^{18}$O measurements. Shell dating and sampling strategies vary among studies (1-345 samples per shell, mean 44.7 samples per shell) and vary significantly by journal discipline. Data are from various bivalves and gastropod species, with *Mytilus* spp. being the most commonly analyzed taxon. This novel database can be used to investigate changes in nearshore sea surface conditions including warm-cool oscillations, heat waves, and upwelling intensity, and provides nearshore calcium carbonate $\delta^{13}$C and $\delta^{18}$O values that can be compared to the vast collections of offshore foraminiferal calcium carbonate $\delta^{13}$C and $\delta^{18}$O data from marine sediment cores. By utilizing previously published geochemical data from midden and museum shells rather than sampling new specimens, future scientific research can reduce or omit the alteration or destruction of culturally valued specimens and sites. The data set is publicly available through PANGAEA at https://doi.org/10.1594/PANGAEA.941373 (Palmer et al., 2021).

# 1 Introduction

## 1.1 Nearshore records of environmental change

Analysis of past climatic and oceanographic change is critical for understanding scales of natural variability in marine systems and is thus essential to predict outcomes of modern climate change. Paleoceanographic studies largely focus on offshore marine sediment archives because they provide long-term, continuous archives of ocean and climate change over time (e.g. Hendy et al., 2002; Lisiecki and Raymo, 2005; Schöne and Gillikin, 2013; Gillikin et al., 2019). Yet, we lack similar continuous records of nearshore environments, preventing quantification of past oceanographic changes in these systems. Shell material from nearshore marine molluscs are archives of the environment in which these organisms lived, and shifts in oxygen and carbon isotope ratios indicate changes in sea surface temperature, salinity, and upwelling (Andrus, 2011). Additionally, such archives can provide insight into the role of nearshore marine molluscs in ecosystems and as resources for human communities (Thomas, 2015a, b; Twaddle et al., 2016). In the coastal California Current System, an abundance of sub-fossil marine mollusc shells are found in archaeological middens, which have been analyzed to provide environmental proxies and to understand historical food sources and trading routes (Rick et al., 2006a; Braje et al., 2012; Burchell et al., 2013a, b; Eerkens et al., 2013; Cannon and Burchell, 2017). Nearshore shell archives typically represent snapshots of paleoceanographic history due to the limited lifespan of these organisms and inconsistencies in the preservation of these archaeological and geologic records compared to long-term offshore marine records (Kennett and Kennett, 2000; Rick et al., 2006a; Robbins and Rick, 2006; Andrus, 2011). Here we synthesize previously published records of oxygen and carbon isotopes in nearshore marine molluscs in order to provide an archive that allows researchers to ask questions across broader spatial and temporal scales. We address the following questions in this paper to provide context for our database:

1) What are the spatial, temporal, and taxonomic distributions of previously collected geochemical data from nearshore marine molluscs from the California Current System?
2) What metadata exists for these shell geochemistry records (taxa, age, latitude and longitude)?
3) What strategies are used to sample individual shells? How do sampling strategies differ between studies that collected data for archaeological vs. oceanographic research?

## 1.2 Oxygen and carbon isotopes as proxies for oceanographic reconstruction

Oxygen and carbon isotopes from molluscan shells are established proxies for past environmental and oceanographic change. In an oceanographic context, oxygen isotopes of shell carbonate are used as a proxy for changes in relative sea surface temperature, salinity, and ice volume (Andrus, 2011; Gillikin et al., 2019). In archaeological research, oxygen isotopes analyses of the outermost growth edge of shells (the most recent growth before death) are often used to infer sea surface temperatures that reflect seasonal human foraging patterns (Butler et al., 2019). Shell carbon isotopes are a proxy for the isotopic signature of dissolved inorganic carbon, which is largely dependent on biological productivity and respiration (Butler et al., 2019). *Vital effects,* or offsets from expected isotopic equilibria due to biological processes of each individual species and organism, confound interpretations of oxygen and carbon isotopes as oceanographic proxies (Gröcke and Gillikin, 2008). The metabolic

rate, calcification rate, growth mechanisms, ontogenetic patterns, and site of calcification within the organism can all lead to offsets between environmental and shell carbon and oxygen stable isotope values (Gillikin et al., 2006; McConnaughey and Gillikin, 2008; Ford et al., 2010; Schöne and Gillikin, 2013). Here, we do not attempt to adjust values for the vital effects of each species or to convert carbonate oxygen or carbon isotope values into environmental variables (e.g. sea surface temperature), rather we compile the raw values as reported in each study to maintain consistency within the database.

**1.3 Archaeological shells as environmental archives**

Shell middens - the remnants of harvesting by Indigenous humans - are common along the west coast of North America. These middens are composed of diverse assemblages of intertidal and subtidal invertebrate taxa, including arthropods, bivalves, echinoderms, and gastropods. Along the western North American coast, evidence for human occupation and human-environment interactions in the nearshore environment extends back to the late Pleistocene (Rick et al., 2006a; Andrus, 2011; Becerra-Valdivia and Higham, 2020). Well preserved archaeological records from the California Channel Islands show evidence of human communities that relied heavily on marine hunting, fishing, and foraging throughout the Holocene (Rick et al., 2005). Marine molluscs were an important dietary and cultural resource for many Native peoples living along the North American coast through the Holocene (Jones and Richman 1995, Braje et al., 2012, Vellanoweth et al., 2006, Jones and Richman 2012). Archaeological shells are ideal candidates for paleoceanographic analysis in this region because they are relatively abundant along the North American west coast and many shells have been isotopically analyzed for environmental reconstruction in past studies.

Synthesizing previous work conducted by both archeologists and paleoceanographers in the context of environmental reconstruction will allow for the examination of a substantial amount of existing nearshore data while minimizing future destructive sampling. Shell middens remain important cultural sites for many Indigenous tribes today; as such, reducing or eliminating the use of additional destructive testing (including isotopic analysis) for scientific analysis maintains the cultural value of middens and respects cultural practices and traditions of tribes. Further, specimens maintained in museum collections are often not available for destructive analysis. Thus, by synthesizing previously published work and by increasing collaboration and conversations between archeologists, geochemists, and paleoceanographers, we can gain further insight across all fields without conducting additional destructive sampling.

**1.4 Shell Sampling Strategies**

Paleoceanographers and archaeologists target different questions, and thus utilize divergent sampling strategies when analyzing the geochemistry of nearshore mollusc shells. Small, often powdered, samples of calcium carbonate are extracted from the whole shell for analysis. Subsampling, here defined as taking more than one measurement from the same shell, can be achieved through drilling with a micromill or handheld drilling device (such as Dremel tool). The number of subsamples collected from a given shell and the location of the subsampling varies broadly

across disciplines and individual researchers. Archeologists who seek to determine the seasonality of  shell

collection to better understand human history often conduct isotopic analyses only of the outermost edge of a shell

(Eerkens et al., 2013; Jazwa and Kennett, 2016). This may involve a single sample or multiple subsamples from an

individual shell taken at the terminal margin of the shell to capture conditions near the time of collection (Andrus,

2011). Alternatively, researchers seeking to use shells as an archive of oceanographic change through time or to

understand the life history of the mollusc may use higher resolution sequential sampling (high sampling density,

multiple closely spaced samples) in order to generate a record of isotopic changes over time (Takesue and van Geen,

2004; Ferguson et al., 2013; Robbins et al., 2013). For this approach, a micromill or other drilling tool is typically

used to sequentially sample a shell at multiple points along a growth (time) axis (Andrus, 2011). The new database

presented here provides opportunities to examine the variation in shell sampling protocols used among researchers

from different disciplines and explore how these methods may limit or enhance the usefulness of sampling at

different resolutions for paleoceanographic interpretations.

## 2 Methodology

Here, we assembled a database of oxygen and carbon isotope values from nearshore marine molluscs extracted from

peer-reviewed publications in archaeological, geochemical, and oceanographic journals. The database is structured

with each entry (row) representing a unique geochemical measurement; multiple subsamples were often collected

from individual shells; therefore a single shell may have multiple entries (rows) for each subsample drilled along the

shell. Detailed metadata were recorded for each individual data point including, when available: paper of original

publication, publication year, sample number (given by original authors), subsample number (given by original

authors), age in years before present, age range (if original authors reported age as a range of values for a

stratigraphic section), species, source (midden or modern), latitude, longitude, calculated sea surface temperature

(only if published by original authors), archaeological trinomial (when applicable). For every entry in the database,

we added a unique shell identification number for each individual and a unique subsample identification number

(when applicable). We also recorded the number of subsamples per shell, which is the number of calcium carbonate

samples extracted from and analyzed for oxygen or carbon isotopes for each given shell.

Additionally, we compiled the following metadata about the species represented in the database: tidal height

(intertidal, subtidal), life mode (infaunal, epifaunal), and habitat (estuary, open coast). Life mode represents the

faunal niche of the species, either infaunal or epifaunal (Table 1). Habitat indicates the marine environment as

determined by geographic location, proximity to coast, species, and reported environment in the original publication

(Table 1).

| Species | Common Name | Tidal Height | Life Mode | Citations |
|---|---|---|---|---|
| *Chione fluctifraga* | | intertidal | infaunal | (Keen, 1971) |
| *Chione cortezi* | | intertidal | infaunal | (Keen, 1971) |

| *Haliotis cracherodii* | Black abalone | intertidal | epifaunal | (Tissot, 1988; Light et al., 2007) |
|---|---|---|---|---|
| *Haliotis rufescens* | Red abalone | intertidal, subtidal | epifaunal | (Díaz et al., 2000; Light et al., 2007) |
| *Macoma* sp. | | intertidal | infaunal | (Light et al., 2007) |
| *Crassostrea gigas* | Pacific oyster (Magallana gigas) | intertidal | epifaunal | (Light et al., 2007) |
| *Mytilus californianus* | California mussel | intertidal | epifaunal | (Suchanek, 1978, 1981; Light et al., 2007) |
| *Mytilus edulis* | Blue mussel | intertidal | epifaunal | (Suchanek, 1978, 1981; Light et al., 2007) |
| *Mytilus* sp. | Mussel | intertidal | epifaunal | (Seed and Suchanek, 1992; Light et al., 2007) |
| *Mytilus trossulus* | Bay mussel | intertidal | epifaunal | (Seed and Suchanek, 1992; Light et al., 2007) |
| *Olivella biplicata* | Purple olivella | intertidal, subtidal | infaunal | (Light et al., 2007) |
| *Panopea abrupta* | Geoduck | Intertidal, subtidal | infaunal | (Light et al., 2007) |
| *Prototoca staminea* | Littleneck clam | Intertidal, subtidal | infaunal | (Fraser and Smith, 1928; Takesue and van Geen, 2004; Light et al., 2007) |
| *Saxidomus gigantea* | Butter clam | Intertidal | infaunal | (Hallman et al., 2011) |

**Table 1:** Common name, tidal height, and life mode for all taxonomic groups included in the database.

Age model type varied across publications. If the age of a shell was reported as an age range of a midden section, the age range is included in the database and the midpoint of the reported age range is included in the database as the age in years before present. For all archaeological shells, age in years before present designates 1950 as present, following the convention of radiocarbon dating and most archaeological studies. For modern samples, age in years before present represents age in years before 2020. Thus, for users of the database, a 70-year correction must be applied to align all ages to the same timescale. We adhere to this dual system to maintain the ages as they were presented in each publication and to prevent negative values for age in years before present. All oxygen and carbon isotope data published in the original papers are included. Oxygen and carbon isotope values are reported in per mil (‰) relative to Vienna Pee Dee Belemnite (VPDB). In one instance, we included data in the database that the author reports as unreliable (see Flores 2017 for full explanation of data quality).

After compilation of the database, we quantified the geographic distribution of the data, temporal distribution of the data, number of taxa, relative abundance of taxa represented in the database, range of subsamples within an individual shell, and range of $\delta^{13}C$ and $\delta^{18}O$ values. Differences in $\delta^{18}O$ and $\delta^{13}C$ among species and in the subsamples per shell by journal disciplines were quantified using ANOVA and Tukey's Test using R (R Core Team, 2021). The database is publicly available through PANGAEA at https://doi.org/10.1594/PANGAEA.941373 (Palmer et al., 2021). Data publication adheres to FAIR Principles.

## 3 Results and discussion

### 3.1 Data Sources

We identified and included 29 previously published studies in the database. Our aim was to create a database that could be used for paleoenvironmental reconstruction at various temporal scales, as such, we included published datasets that reported isotopic data within the context of shell growth, i.e., the directionality and therefore seasonality of isotopic patterns in time could be determined. For this reason, isotopic analyses of fragments or broken shells, often for archaeological studies, are not included in the database. Studies included in the database were published in archaeological (14), geochemical (5), and paleoclimate and paleoceanography journals (10). The database includes 598 shells including 4917 carbon isotope values and 7366 oxygen isotope values. Data was published from 1994-2021; the number of papers published increases towards the present. Data was largely sourced from midden specimens (516) and to a lesser extent from modern collected specimens (81).

### 3.2 Spatial distribution of data

The database includes analyses of shells collected from 30ºN to 55ºN along the western coast of North America. The spatial distribution of data is not continuous, due to the nature of human settlement and differences in the preservation and sampling of middens across space (Fig. 1). More data are available from California relative to Oregon, Washington, or British Columbia (Fig. 1). A significant portion of the available data are from midden shells collected on the northern Channel Islands, California due to exceptional preservation in a semi-arid climate (Fig. 1) (Rick et al., 2006b). Based on known environmental associations of species and location of midden sites, we determined whether specimens were derived from open coast or estuarine sites. Most studies analyzed shells collected from open coast sites and a few studies analyzed specimens from estuarine environments (Fig. 4E, F). We acknowledge that specimens may have been moved long distances by those harvesting and consuming them, thus creating a compounding variable for further analysis. Due to variability in original sampling sites and potential of over-land transport by gatherers, it is challenging to determine the geographic origin of each specimen. Further, without constraining past water oxygen isotope values (influenced by salinity), it is difficult to estimate paleotemperatures using these data.

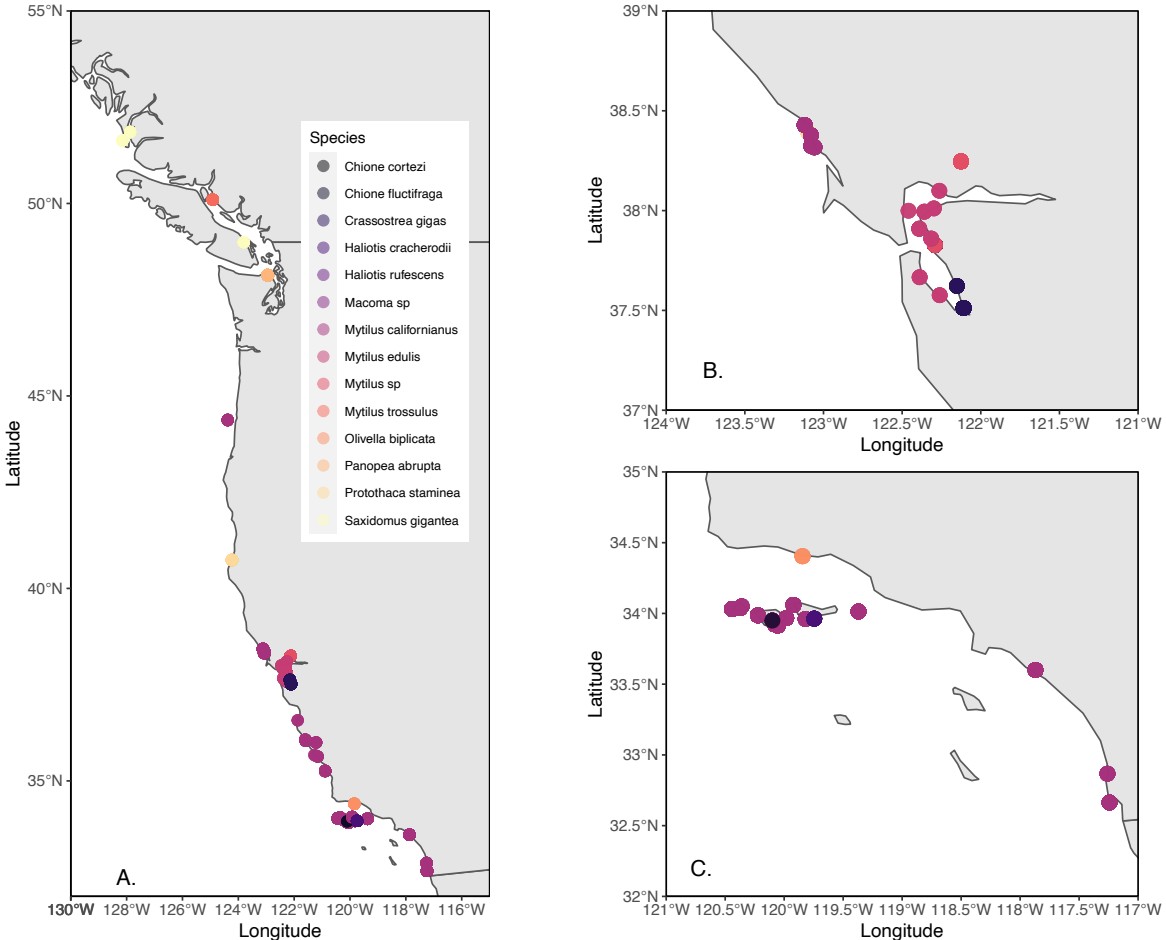

**Figure 1:** Maps of location of collection (for modern) and midden sites (for midden) of all specimens included in database. All specimens included in map (A.), zoom in on northern California (B.) and southern California (C.).

### 3.3 Temporal distribution of data

The age range of the shells in the database is 8800 - 0 BP (Fig. 2). The methodology for age control on individual shells varied among studies; some provide age ranges defined by radiocarbon dating of material that was co-collected with shells from stratigraphic sections of the midden, some studies provide radiocarbon dates from individual shells, and some shells in the database were collected live. Eleven papers reported age ranges: Burchell et al., 2013, Eerkens et al., 2005, Eerkens et al., 2013, Eerkens et al., 2016, Glassow et al., 1994, Glassow et al., 2012, Hallman et al., 2013, Jazwa and Kennett 2016, Jones and Kennett 1999, Rick et al., 2006, and Robbins and Rick 2006. In cases when age was reported as a range, the range is listed in the database (age_range column) and the midpoint of the range is recorded as the age of shell (age_ybp). The majority of entries in the database are from midden shells (4857) compared to modern shells (2553). The data are biased towards the present; more data are available from 0-3000 BP (4718 entries) relative to 3000-9000 BP (2192 entries).

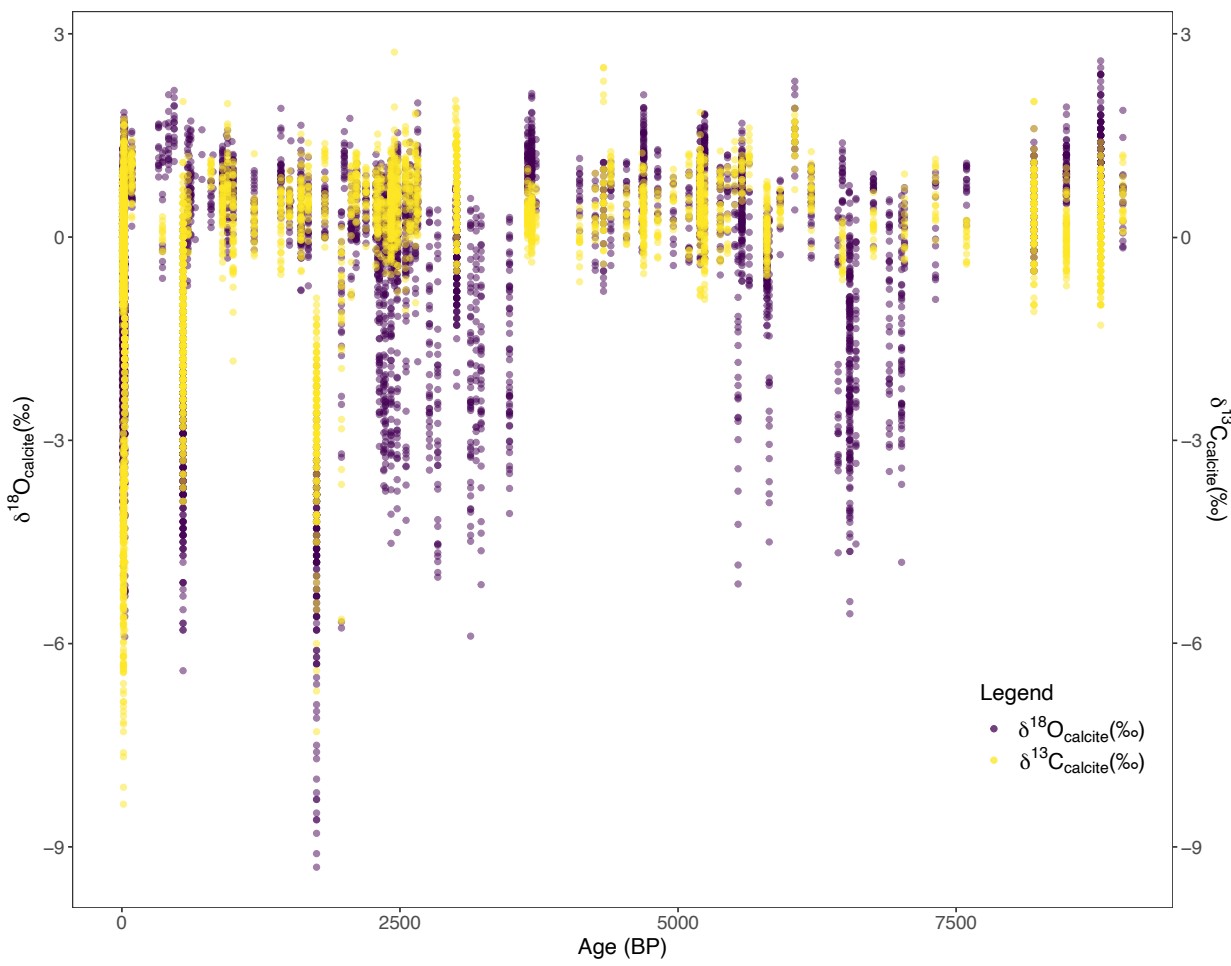

**Figure 2:** Oxygen (purple) and carbon (yellow) isotope values (‰VPDB) through time in years before present.

### 3.4 Taxonomic distribution of data

Available data represent 14 taxa: *Crassostrea gigas, Chione cortezi, Chione fluctifrage, Haliotis cracherodii, Haliotis rufescens, Macoma* sp., *Mytilus californianus, Mytilus edulis, Mytilus* sp., *Mytilus trossulus, Olivella biplicata, Panopea abrupta, Protothaca staminea, and Saxidomus gigantea* (Fig 3)*.* The available data and thus the database is largely biased to *Mytilus* taxa (3739 entries for *M. californianus,* 138 for *M. edulis,* 53 for *M. trossulus,* 464 for *Mytilus* sp.,). This finding is in agreement with previous work showing that *Mytilus* spp. are highly abundant in coastal Western North American middens throughout the Holocene as *Mytilus* spp. are a common food choice of Indigenous peoples in this region (Jones and Richman, 1995; Braje et al., 2012; Thakar et al., 2017).

Oxygen and carbon isotope values vary significantly between taxa groups when analyzing all data across the entire age range of the data. Results of ANOVA indicate that there are significant differences between taxa in regards to oxygen isotope values (ANOVA, $p < 0.05$). The Tukey post hoc test revealed six groupings of taxa that are statistically similar to one another ($p < 0.05$): Group a: *Haliotis rufescens*, *Olivella biplicata*, and *Panopea abrupta*, Group b: *Olivella biplicata*, *Panopea abrupta,* and *Protothaca staminea,* Group c: *Chione fluctifraga*, *Haliotis*

*cracherodii*, *Mytilus californianus*, *Olivella biplicata*, and *Protothaca staminea,* Group d: *Chione cortezi*, *Chione fluctifraga, Mytilus edulis,* and *Saxidomus gigantea,* Group e: *Crassostrea gigas, Macoma sp.,* and *Mytilus trossulus,* Group f: *Mytilus sp.* (Figure 3). Results of ANOVA indicate that there are significant differences between taxa in regards to carbon isotope values (ANOVA, p<0.05). The Tukey post hoc test revealed seven groupings of taxa that are statistically similar to one another (P<0.05): Group a: *Haliotis rufescens*, *Olivella biplicata*, and *Panopea abrupta*, Group b: *Haliotis cracherodii* and *Panopea abrupta,* , Group c: *Chione cortezi*, *Chione fluctifraga*, and *Mytilus californianus*, Group d: *Chione cortezi*, *Chione fluctifraga, Macoma sp.*, *Protothaca staminea*, and *Saxidomus gigantea,* Group e: *Mytilus* sp.,  Group f: *Mytilus edulis*, and Group g: *Crassostrea gigas* (Fig. 3).

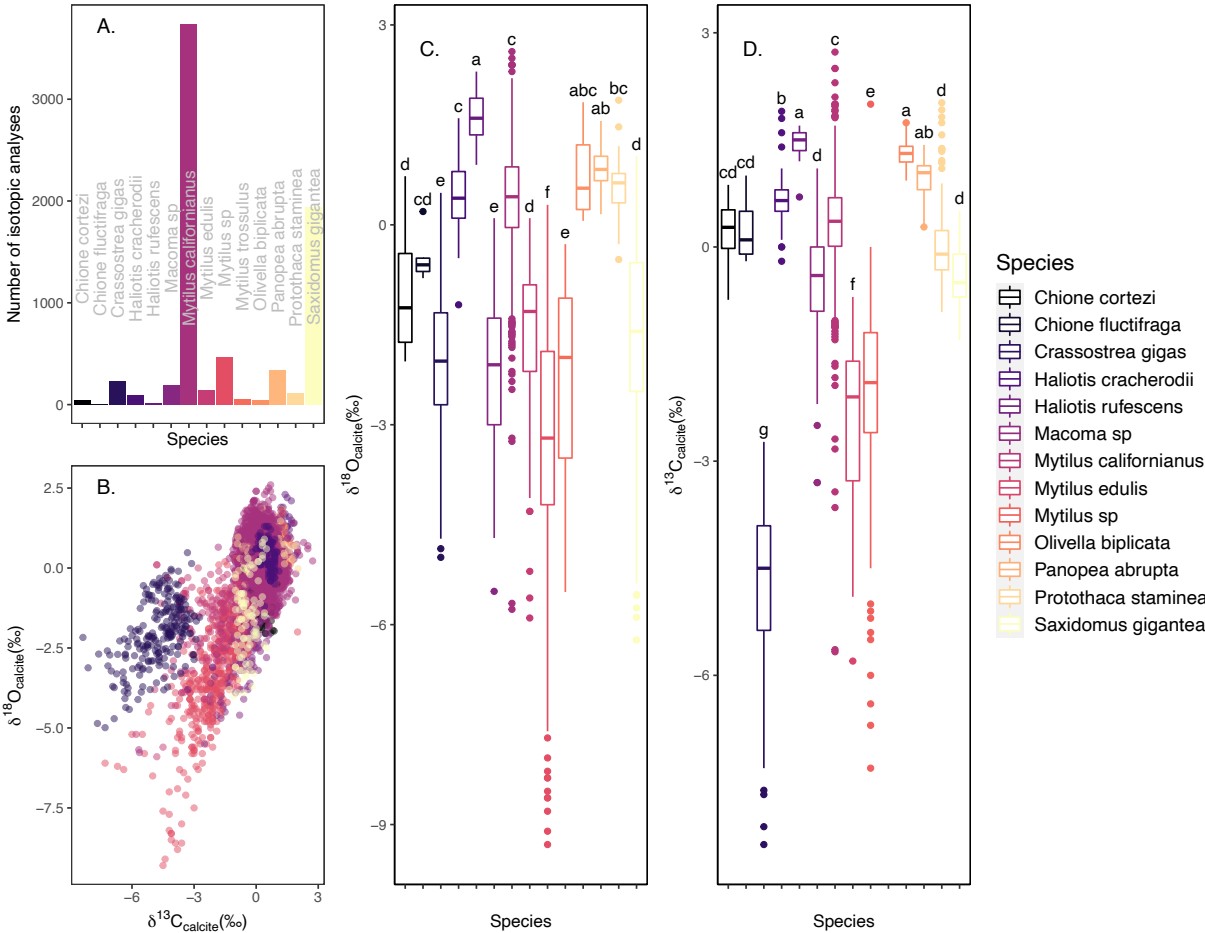

**Figure 3:** Taxonomic distribution of data. Histogram of number of entries in the database by taxonomic group (A.) Carbon vs. oxygen isotope values (‰ VPDB) colored by taxonomic group (B.). Box plot showing oxygen (C.) and carbon (D.) isotope values by taxonomic group. Letters in panels C and D indicate which taxa are statistically similar in terms of oxygen and carbon isotope values, respectively (Tukey, p>0.05).

### 3.5 Ecological distribution of data

The addition of several ecological traits expands the breadth of research topics that can be explored using the database. Nearly all the species are intertidal and a few species (*Olivella biplicata*, *Haliotis rufescens, Panopea abrupta, Protothaca staminea*) are known to be intertidal and subtidal (Fig. 4). Most specimens are from taxonomic groups with epifaunal life modes (497), and some have infaunal life modes (100) (Fig. 4). Both tidal height and life mode impact organismal exposure to air, seawater, and porewater and thus can influence oxygen and carbon isotope values. Species and populational variation in latitude and tidal height impacts exposure to terrestrial freshwater influences, which can also influence oxygen and carbon stable isotope values. As such, future researchers may consider isolating specimens with certain life modes or tidal heights to address specific questions in paleoclimate reconstruction. Additionally, inter and intra species variability further complicate the ability to make cross site and cross species comparisons without constraining $\delta^{13}C$ and $\delta^{18}O$ of seawater.

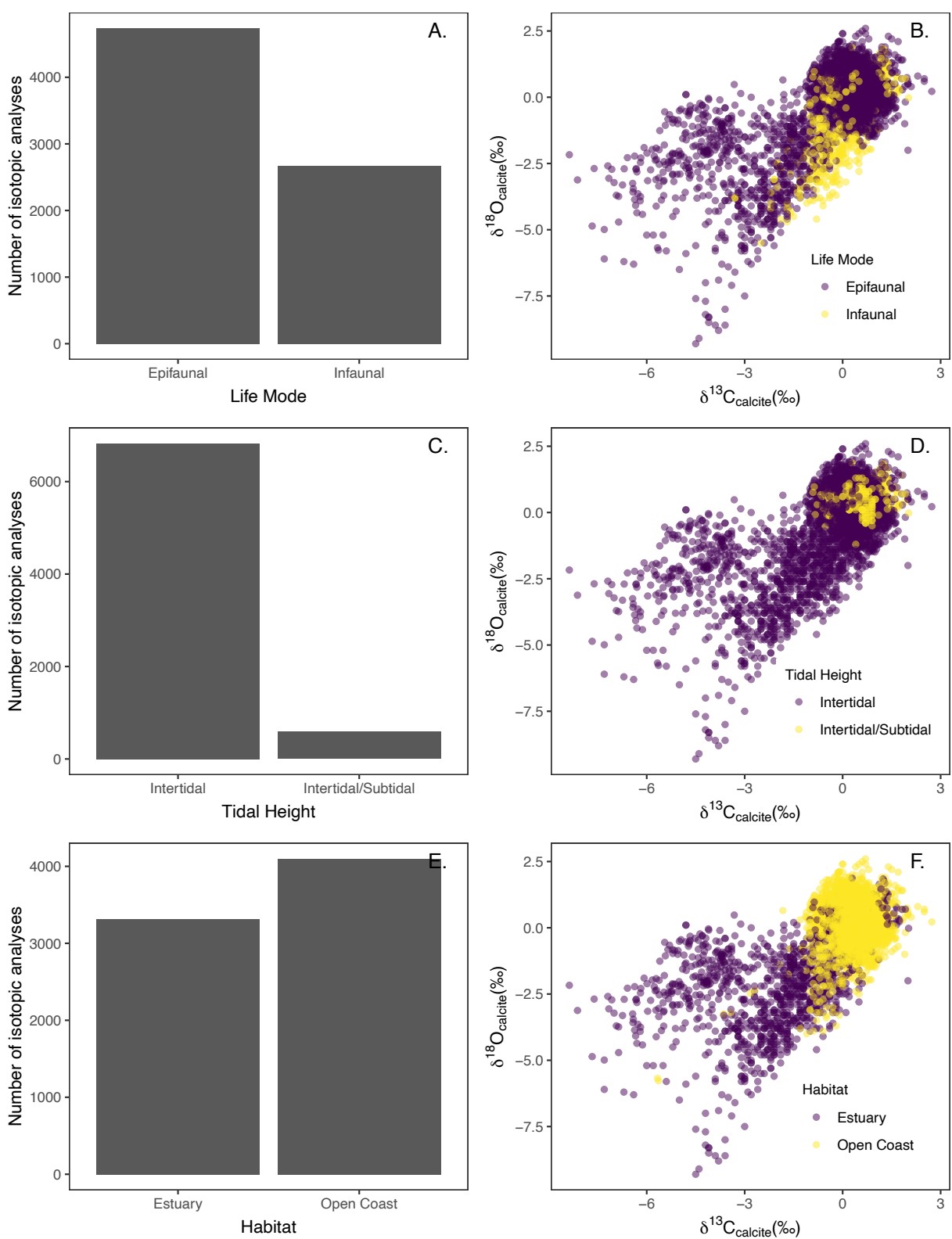

**Figure 4:** Ecological distribution of data. Histogram and oxygen vs. carbon isotopes are respectively colored by category for life mode (A., B.), tidal height (C., D.), and habitat (E., F.).

### 3.6 Variation in sampling strategy

The sampling strategy for obtaining carbonate for isotopic analysis differed widely among studies and within studies with important implications for data analysis. In some studies, a single sample was obtained from the outermost growth edge of the shell (Fig. 5). We document a wide range of subsampling resolution, from 1 to 345 subsamples

from a single shell (Fig. 5). The number of subsamples reported per shell varies significantly among journal types (ANOVA, $p < 0.05$) paleoclimate and paleoceanography studies utilized the most subsamples (mean = 61.9 subsamples per shell), followed by geochemical studies (mean = 38.1 subsamples per shell), and archaeology papers utilized the fewest subsamples (mean = 14.4 subsamples per shell) (Fig 5). The variation in approach and methodology in these studies impacts the resolution of data available in the database. Additionally, changes in shell

growth rate over time and shell density may impact the temporal resolution of data sampled at regular intervals from a single shell and, in turn, influence environmental interpretations (de Winter et al., 2021; Vriesman et al., 2022). Variation in shell growth rate within single shells or between shells will also affect the degree to which individual samples are time averaged and merits further exploration. Future research can explore how multiple approaches to subsampling impacts how informative the oxygen and carbon isotope values from each shell are for use in

environmental reconstruction. Additionally, the diversity in sampling type motivates future interdisciplinary collaborations that can answer multiple research questions using the same archive. For example, single shells could be utilized to answer questions about both archaeological histories and past oceanographic conditions.

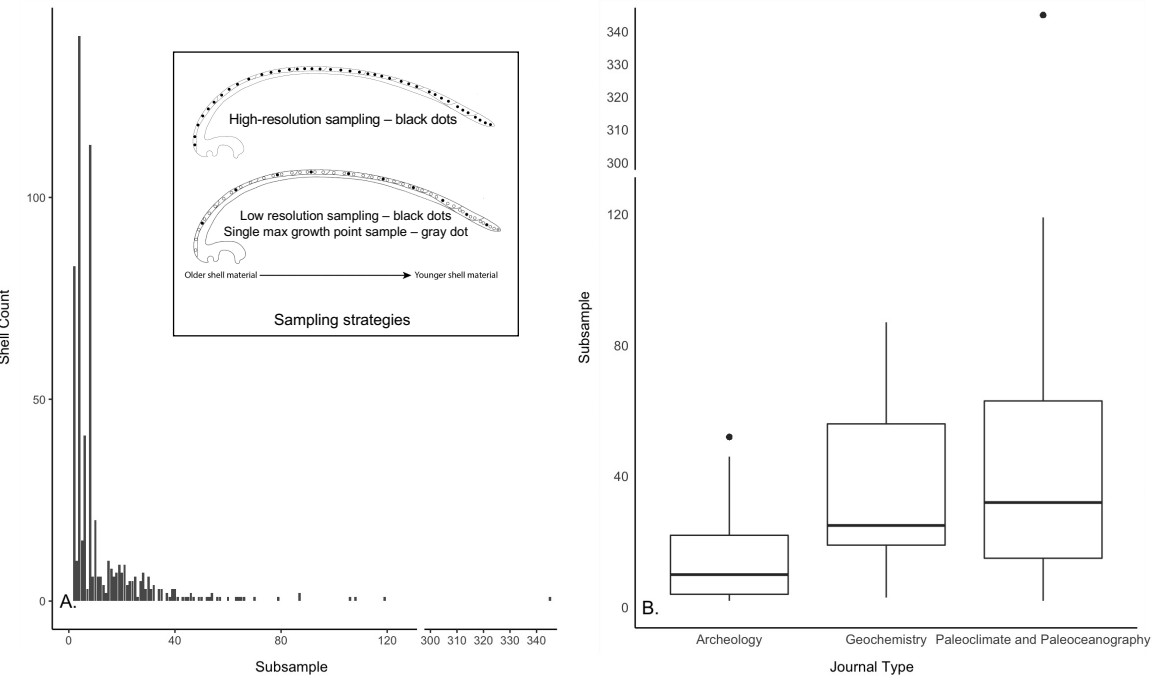

**Figure 5:** Distribution of occurrence of number of subsamples from an individual shell (A.) Panel A inset is a

schematic of sampling strategies showing multiple degrees of sampling resolution. Box and whisker plot of subsamples per shell by journal type (B.), difference between each group is significant (ANOVA, Tukey, $p < 0.05$).

**4 Data Availability**

The database described here is available at https://doi.org/10.1594/PANGAEA.941373 (Palmer et al., 2021). The
dataset is subject to a Creative Commons Attribution 4.0 International license agreement (CC-BY-4.0). In the
published database, each row represents an individual geochemical analysis, and multiple rows of data are often
from the same shell. Detailed metadata are recorded for each individual data point including, when available: shell
identification number (generated here), subsample identification number (generated here), number of subsamples
per shell, name of publication (author, year), name of sample from original publication, name of subsample from
original publication, latitude, longitude, $\delta^{13}$C, $\delta^{18}$O, age in years BP, age range (if reported in paper), species, source
(modern, archive, burial, midden), suggested $\delta^{18}$O correction published in the original publication, reconstructed sea
surface temperature if reported in original paper, archaeological trinomial (if reported), tidal height of organism, life
mode of organism, habitat type of organism, year of publication, and journal type (archaeology, geochemistry,
paleoclimate and paleoceanography).

**5 Conclusions**

Marine mollusc geochemical data provides a record that is needed to understand changes in nearshore environments
through time. In the California Current System, the abundance of previously published data on archaeological,
archival, and modern specimens allows for exploration of oceanographic and climate change through the Holocene.
Given the utility of archaeological records to provide long-term environmental archives, we encourage the use of
these datasets in future paleoceanographic studies. Further, by synthesizing previously published data we provide an
archive in which future work can be conducted without accessing culturally valued Indigenous artifacts or
conducting destructive analysis on museum specimens. Future work on the data published here can explore changes
in nearshore marine mollusc geochemistry across geographic, environmental, or biological gradients.

**6 Land and Data Acknowledgement**

Midden shells analyzed and synthesized for this paper were originally collected and gathered by Indigenous people
from across the West Coast of North America. We do not attempt to name all tribes whose ancestral and present
homelands make up this study area, but we acknowledge that the majority of the geographic area covered here
represents unceded land of Indigenous tribes, and that data used here from previously published studies may have
been acquired without consent from Indigenous peoples. We direct readers to the open-source resource:
nativelands.ca to identify the homelands of the diverse Indigenous people of this region and encourage readers to
use this resource as a starting place to learn about the land and marine stewardship of Indigenous peoples past and
present.

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
