# Peer review of "Compilation of a database of Holocene nearshore marine mollusc shell geochemistry from the California Current System"

_Earth System Science Data, 2021_

## Author Comment (AC1)

RC1: Reviewer comment (black)
AC: Author response (gray)

AC: We thank Dr. Andrus for their time and contributions to improving the database and manuscript. We have responded to each point of the review below.

**General Comments:**

RC1: Overall, I think this work represents a useful contribution to archaeology and paleoclimatology. The data are from different sources yet form a coherent set. My main overarching comment is the absence of at least one common midden clam found along the coast between Oregon and Alaska; butter clam (Saxidomus gigantea). The authors may have intentionally left it out because of its more northern range, but it does overlap with the database they built. I certainly don't think this omission should prevent publication of this database, and one could argue that the only published isotope data on ancient butter clams are from sites just north of Vancouver Island, thus beyond the scope of this work, but if the authors wish to add it, I could see its relevance.

AC: We will add data from *Saxidomus gigantea* as it was recommended by both reviewers. This change will be made in the database and submitted to Pangea and the manuscript will be updated to include the additional data set.

RC1: The statistical analyses are interesting, and I am glad they are included, but interpreting their meaning will be challenging due to the impact of habitats and water isotope composition on these data. Therefore, I am especially interested in section 3.6 and found the plots illuminating. Much could likely be explained by the position of the ancient collection sites along the salinity gradient, but this is not known in most archaeological contexts (e.g see panel F in Figure 4). It might be useful to again remind the readers that without constraining past water oxygen isotope content, it is difficult to interpret these shell isotope data for paleotemperature estimates. The authors address this in section 1.2, but it might be useful to state the problem of uncontrolled water oxygen isotopes more explicitly.

AC: We will add additional statements in the results section (3) to explicitly restate the problem of unconstrained water isotope values.

RC1: I recommend this be published with the below edits. I do not know if it is possible to revisit and expand these contributions later, but I am aware of several ongoing projects that will add more data relevant to this set in coming years and I hope they can be included too when they are published.
AC: We appreciate your interest in continuing to expand the database and we welcome future additions to the database.

**Specific and Technical Comments:**

RC1: Line 15: The term "Northeast Coast" is used to denote the study area. I understand that the authors are referring to the northeast coast of the overall Pacific Ocean (as In lines 21 and 22), but typically coasts are defined relative to landmasses, so this might be confusing to some readers. It might be clearer to state the northwest Pacific coast of North America.
AC: The text will be updated to reflect the change.

RC1: Line 28: "Calcite" should be "calcium carbonate" because some of the taxa reported are aragonite or contain both calcite and aragonite in different shells layer.
AC: The text will be updated to reflect the change.

RC1: Line 61: Again, I would avoid the use of the term "biocalcite" and replace this with "biocarbonate" since there are two minerals present in this database.
AC: The text will be updated to reflect the change.

RC1: Line 64: I would suggest adding the word "relative" before "sea surface temperatures" since in the cited papers the season of capture is determined independent of absolute temperature calculations.
AC: The text will be updated to reflect the change.

RC1: Lines 125-126: It might not be possible, but if the original data sources provide estimates of time-averaging it might help readers to include that in this database.
AC: In order to maintain consistency across the database we were not able to include a uniform metric of time averaging. The column in the database labeled age_range does report a range of ages for a stratigraphic section if provided by the original authors. This was not previously included at line 125-126; we will add the following text: age range (if original authors reported age as a range of values for a stratigraphic section).

RC1: Table 1: Some of the taxa are listed as intertidal, but they are also found as fully subtidal (e.g. Panopea abrupta, Protothaca staminea). Each listed species should be rechecked to ensure their total depth range is recorded, although the practical depth of shell collection by ancient humans would constrain the maximum reasonable depth of collection.
AC: The text in Table 1 will be updated to reflect the change. Changes to the database reflecting this change will be submitted to Pangea. Figure 4 will be updated. Section 3.5 will be updated.

RC1: Line 149: "13" should be in superscript.
AC: The text will be updated to reflect the change.

RC1: Section 3.5: A related concern to sampling density is the growth rate of the shell. A densely sampled, slow growing, shell may still have long time averaging, whereas a low-density sampling of a fast growing shell may yield finer temporal averaging. It is beyond the scope of this paper to parse out all possible such patterns, but if the original papers include such data it might be useful to reference it, to at least remind the reader of this concern.
AC: We agree this is critical and will add two additional sentences to remind the reader of this concern.

---

## Author Comment (AC2)

**Response to RC3**

RC3: Reviewer comment (black)
AC: Author response (gray)

RC3: I very much agree with this reviewer that it could be useful to examine estimates of time-averaging where available or at least remind the readers that time-averaging will vary depending on sampling density and shell growth rate.
AC: We agree this is critical and we will add two additional sentences to remind the reader of this concern.

---

## Author Comment (AC3)

**Response to RC2**

RC2: Reviewer comment (black)
AC: Author response (gray)

AC: We thank the reviewer for their time and contributions to improving the database and manuscript. We have responded to each point of the review below.

RC2: Overall, I think this work addresses an important and recurring challenge with paleoclimate/ocean datasets garnered from archaeological samples. This is a nice first attempt to synthesize a wide range of species, ecologies, time periods, and sampling strategies, which is no easy feat!

My main concern is that the geographic designation (Northeast Pacific) of the study area is far broader than the data represented within this manuscript. Most of the data is concentrated along the California coastline. While I do think there may be a greater density of isotopic work on shellfish from California middens, the authors chose to include a handful of data from species found along the coastlines of Washington and southern British Columbia. Therefore, I do think it is important that they also include work that has been done with Saxidomus gigantea (both modern and archaeological) in Washington and British Columbia. Moreover, the usage of Northeast Pacific does imply the coastline of the North American continent, leaving me questioning if it appropriate to exclude work done in the Gulf of Alaska (Hallmann et al., 2009, 2011, 2013; Bassett et al., 2019). To summarize, if the authors wish to include their current data from Washington and southern British Columbia, they also need to include work done with Saxidomus gigantea and if they are truly considering the Northeast Pacific, this does also include Alaska (although I think it's perfectly acceptable to omit work in Alaska and more precisely define their geographic focus instead).

AC: We will refine the terminology to define the scope of the study to the "California Current System" rather than the "Northeast Pacific." As such, we will add data from *Saxidomus gigantea*, yet we choose to omit work from Alaska as it is outside the scope of the study.

RC2: My other comment is that I do not think the discussion of how changes in d18O(water) makes direct cross-site and even cross-species comparisons difficult, if not impossible. This is especially true for SST reconstructions from d18O(shell). The locations of the habits of each of these species varies considerably and so too then does their exposure to terrestrial freshwater influences, which is difficult to constrain in archaeological studies. I think section labeled 3.6 (the sub-headers 3.5 and 3.6 are out of order) could be greatly strengthened by detailing this challenge more thoroughly.

AC: We will correct the numbering error of the two sections, and we will add further discussion of these two points in the text.

RC2: Line 28: Is the comosition calcite for all of the species include in this study?
AC: We will update the text to say "calcium carbonate."

RC2: Line 43: Recommend including additional citations - Schone and Gillikin, 2013; Gillikin et al., 2019

AC: We will add recommended citations.

RC2: Line 45: Recommend including additional citation for "...as resources for human communities" - Thomas, 2015a, 2015b; Twaddle et al. 2016

AC: We will add recommended citations.

RC2: Line 47: These citations are only relevant to the California coastline, there are citations for work in southern B.C. as well that should be included since data is included from this region. See Burchell et al. 2013a, 2013b, Cannon and Burchell, 2017. Depending on the final determination of the geographic span of this article, there are several other citations that could be include for northern B.C. and the Gulf of Alaska.

AC: We will add recommended citations. We do not include a discussion of the Gulf of Alaska.

RC2: Line 61: "Biocalcite," again, do all of the species featured in this study produce only calcitic shells?

AC: We will update text to say "shells" to be inclusive of all shell types.

RC2: Line 62-3: "... as a proxy for changes in SST, salinity, and ice volume" needs citation.

AC: We will add recommended citations.

RC2: Line 67-8: Yes, this is correct. I would suggest fleshing out in more detail how biological processes affect oxygen and carbon isotopic values.

AC: We will add an additional sentence describing vital effects. "The metabolic rate, calcification rate, growth mechanisms, ontogenetic patterns, and site of calcification within the organism can all lead to offsets between environmental and shell carbon and oxygen stable isotope values (Ford et al., 2010, Gillikin et al., 2006, McConnaughey and Gillikin 2008, Schone and Gillikin, 2013).

RC2: Line 105-107: It is unclear how the categories of "multiple subsamples" and "higher resolution sequential sampling" are distinguished from one another. Are you defining these groups based on #samples/annuli? Overall time-averaged?

AC: We will clarify this in text. To clarify here, 'higher resolution sequential sampling' refers to high sample density, or lots of closely spaced samples to capture as much environmental variation as possible. 'Multiple subsamples' simply refers to more than one subsample taken at the terminal margin of the shell to capture conditions closer to the time of collection"

RC2: Line 122: It is unclear to me how you selected data from modern samples. Are they only from museum collections? If you are generally considering live-collected modern samples, there are certainly data from the PNW region as well (see map in Bassett et al., 2019 and citations therein for review of these modern data points).

AC: We aim to be inclusive of modern data and we will add in the recommended datasets and submit the changes to the Pangea database.

RC2: Line 170-1: Again, there's actually more data from southern Washington and B.C. that could be included. See suggested citations from Line 47.

AC: We will add in the data as suggested.

RC2: Line 173-4: Authors state that most of the studies included in the present study analyzed shells collected from open coast sites and few studies analyzed estuarine species. Are there no estuarine species present in middens at any of these sites? Certainly middens in Washington state include one estuarine species (Saxidomus gigantea). Is the lack of estuarine species in these datasets some kind of sampling bias or an acknowledgement of the difficulty of interpreting SST records from estuarine species?

AC: We use estuarine to refer to the site of the midden, rather than as an ecological designation. Further, we made some determinations about the environment based on the species (e.g., we know that *M. californianus* shells were harvested from open coast sites since this species is found exclusively in rocky intertidal environments with heavy wave disturbance and high salinity). We will clarify this point in the text and include a caveat addressing your point that species may have been moved large distances by those harvesting and consuming them.

RC2: Line 225: Yes, and I would imagine this is likely a result in shellfish collection technology/methods employed by past inhabitants.

AC: We agree.

RC2: Line 276: Very much appreciate the inclusion of land and data acknowledgement!

AC: Thank you.

RC2: Figure 1: Some of the greens are quite difficult to distinguish from one another and in black in white it would be impossible (also important to note that this color gradient is not very accessible to colorblind readers).

AC: The Viridis palette (R) used here is a recommended colorblind-friendly palette. We will examine alternative colorblind-friendly palettes (e.g., magma palette in the Viridis package).

RC2: Figure 3: Species names are very difficult to read (in fact, in print they are impossible to read). I do think the color gradient works much better here than on the map, where the greens are difficult to distinguish from one another.

AC: We will increase the font size in the figure.

RC2: Suggested citations:

Bassett, C., Andrus, C. F. T., and West, C. F. 2019. Implications for measuring seasonality in the marine bivalve, Saxidomus gigantea. Chemical Geology. 10.1016/j.chemgeo.2018.07.004

Burchell, M., Cannon, A., Hallmann, N., Schwarcz, H.P., Schöne, B.R., 2013a. Inter-site variability in the season of shellfish collection on the central coast of British Columbia. J. Archaeol. Sci. 40, 626–636. https://doi.org/10.1016/j.jas.2012.07.002

Burchell, M., Cannon, A., Hallmann, N., Schwarcz, H.P., Schöne, B.R., 2013b. Refining Estimates for the season of shellfish collection on the pacific northwest coast: Applying high-resolution stable oxygen isotope analysis and sclerochronology. Archaeometry 55, 258–276. https://doi.org/10.1111/j.1475-4754.2012.00684.x

Cannon, A. and Burchell, M., 2017. Reconciling oxygen isotope sclerochronology with interpretations of millennia of seasonal shellfish collection on the Pacific Northwest Coast. Quarternary International.10.1016/j.quaint.2016.02.037.

Gillikin, D., Wanamaker, A.D., and Andrus, C. F. T., 2019. Chemical Sclerochronology. Chemical Geology. 10.1016/j.chemgeo.2019.06.016.

Hallmann, N., Burchell, M., Brewster, N., Martindale, A., Schöne, B.R., 2013. Holocene climate and seasonality of shell collection at the Dundas Islands Group, northern British Columbia, Canada-A bivalve sclerochronological approach. Palaeogeogr. Palaeoclimatol. Palaeoecol. 373, 163–172. https://doi.org/10.1016/j.palaeo.2011.12.019

Hallmann, N., Schöne, B.R., Irvine, G. V., Burchell, M., Cokelet, E.D., Hilton, M.R., 2011. an Improved Understanding of the Alaska Coastal Current: the Application of a Bivalve Growth-Temperature Model To Reconstruct Freshwater-Influenced Paleoenvironments. Palaios 26, 346–363. https://doi.org/10.2110/palo.2010.p10-151r

Hallmann, N., Burchell, M., Schöne, B.R., Irvine, G. V., Maxwell, D., 2009. High-resolution sclerochronological analysis of the bivalve mollusk Saxidomus gigantea from Alaska and British Columbia: techniques for revealing environmental archives and archaeological seasonality. J. Archaeol. Sci. 36, 2353–2364. https://doi.org/10.1016/j.jas.2009.06.018

Schöne, B.R., Gillikin, D.P., 2013. Unraveling environmental histories from skeletal diaries - Advances in sclerochronology. Palaeogeogr. Palaeoclimatol. Palaeoecol. 373, 1–5. https://doi.org/10.1016/j.palaeo.2012.11.026

Thomas, K.D., 2015a. Molluscs emergent, Part I: Themes and trends in the scientific investigation of mollusc shells as resources for archaeological research. J. Archaeol. Sci. 56, 133–140. https://doi.org/10.1016/j.jas.2015.01.024

Thomas, K.D., 2015b. Molluscs emergent, Part II: Themes and trends in the scientific investigation of molluscs and their shells as past human resources. J. Archaeol. Sci. 56, 159–167. https://doi.org/10.1016/j.jas.2015.01.015

Twaddle, R.W., Ulm, S., Hinton, J., Wurster, C.M., Bird, M.I., 2016. Sclerochronological analysis of archaeological mollusc assemblages: methods, applications and future prospects. Archaeol. Anthropol. Sci. 8, 359–379. https://doi.org/10.1007/s12520-015-0228-5

AC: We thank the reviewer for the inclusion of the full citations. We will include all recommended citations.